# Elevating density functional theory to chemical accuracy for water simulations through a density-corrected many-body formalism

Saswata Dasgupta[1,6], Eleftherios Lambros[1,6], John P. Perdew [2,3] & Francesco Paesani [1,4,5✉]

Density functional theory (DFT) has been extensively used to model the properties of water. Albeit maintaining a good balance between accuracy and efficiency, no density functional has so far achieved the degree of accuracy necessary to correctly predict the properties of water across the entire phase diagram. Here, we present density-corrected SCAN (DC-SCAN) calculations for water which, minimizing density-driven errors, elevate the accuracy of the SCAN functional to that of "gold standard" coupled-cluster theory. Building upon the accuracy of DC-SCAN within a many-body formalism, we introduce a data-driven many-body potential energy function, MB-SCAN(DC), that quantitatively reproduces coupled cluster reference values for interaction, binding, and individual many-body energies of water clusters. Importantly, molecular dynamics simulations carried out with MB-SCAN(DC) also reproduce the properties of liquid water, which thus demonstrates that MB-SCAN(DC) is effectively the first DFT-based model that correctly describes water from the gas to the liquid phase.

[1] Department of Chemistry and Biochemistry, University of California, San Diego, La Jolla, CA 92093, USA. [2] Department of Physics, Temple University, Philadelphia, PA 19122, USA. [3] Department of Chemistry, Temple University, Philadelphia, PA 19122, USA. [4] Materials Science and Engineering, University of California, San Diego, La Jolla, CA 92093, USA. [5] San Diego Supercomputer Center, University of California, San Diego, La Jolla, CA 92093, USA. [6]These authors contributed equally: Saswata Dasgupta, Eleftherios Lambros. ✉email: fpaesani@ucsd.edu

ts anomalous behavior[1] and importance to life[2] make water one of the most studied chemical compounds. Among its many unique properties is the high value of the heat capacity that allows water to resist sudden temperature changes, thus permitting living organisms to survive without experiencing significant temperature fluctuations[3]. In addition, the dynamic nature of the water hydrogen-bond network plays a central role in several fundamental processes, including transport and diffusion in bulk solutions and at interfaces, and hydration of hydrophobic and hydrophilic solutes[4]. For example, protein folding is thought to be driven by the hydrophobic effect[5]. Finally, countless chemical reactions involving charged species take place efficiently in liquid water due to its high dielectric constant[6–11]. It is thus not surprising that a myriad of simulation studies have been devoted to developing a fundamental understanding of both chemical and physical properties of water in different environments and under different thermodynamic conditions[12–14].

Density functional theory (DFT)[15,16] is one the most important tools available to computational chemists and physicists for ab initio simulations of molecular systems in the condensed phase since it offers a good balance between accuracy and computational cost[17,18]. However, as discussed in the "Results" section, the accuracy of a DFT calculation depends upon the accuracy of the underlying exchange-correlation (XC) functional, which allows for recasting the many-body electronic structure problem into a (self-consistent) single-particle problem formulated in terms of the Kohn−Sham equations[19]. The simplest XC functional, the local spin density approximation (LSDA)[19–21], was shown to correctly predict the structure of metallic crystals under pressure[22–24], but was unable to fulfill its promises for water simulations, overestimating the strength of the hydrogen bonds and, consequently, predicting a too packed and overstructured liquid phase[25,26]. These limitations hindered the ability of the LSDA functional to describe the properties of water, even qualitatively.

Climbing the Jacob's ladder of DFT approximations[27], the next generation of XC functionals, which were developed within the generalized gradient approximation (GGA)[28–30], dominated the scene of ab initio simulations of water for a long time, owing to their higher accuracy compared to LDA and affordable computational cost. Initial successes of the GGA functionals included relatively accurate binding energies for various water clusters and a reasonable description of the structure of liquid water[25,31–33]. However, it became soon evident that serendipitous error cancellation was the primary reason behind the apparent accuracy of GGA simulations of liquid water, making the predictive power more accidental than consistent[34–37]. For example, it was found that GGA functionals generally underestimate the density of liquid water, while predicting denser ice phases[38].

The third rung of the Jacob's ladder of DFT approximations includes meta-GGA functionals[39,40] that perform significantly better than both LDA and GGA functionals due to the inclusion of the kinetic energy density. Among them, the strongly constrained and appropriately normed (SCAN) functional has gained particular attention because it satisfies all 17 known exact constraints that can be satisfied by a meta-GGA functional[41]. Without being fitted to any bonded system, SCAN was shown to enable accurate predictions for various properties of molecules and solids[42]. In particular, for molecular dynamics (MD) simulations of liquid water, SCAN was found to outperform its predecessor GGA functionals[43,44]. Importantly, accounting for intermediate-range dispersion interactions, the SCAN functional allows for a more accurate description of the energy differences among water clusters and ice phases[42,44], while, when used in MD simulations, it predicts a density of liquid water which is appreciably closer to the experimental value compared to values obtained with GGA functionals[43].

Despite its relatively higher accuracy, SCAN, as all GGA and meta-GGA functionals, is still prone to density-driven errors (defined in the following section), including self-interaction[45] and delocalization errors[46–49]. It was shown that self-interaction errors in the SCAN functional primarily affect 2-body contributions (defined in the "Results" section) to the interaction energies of water clusters[50]. On the other hand, inclusion of a fraction of Hartree−Fock exchange (also known as exact exchange) in a SCAN hybrid was found to partially reduce density-driven errors in the calculations for various water systems[51,52]. However, it was found that increasing the fraction of Hartree−Fock exchange beyond 15% did not improve the accuracy of hybrid SCAN functionals, progressively shifting the structure of liquid water towards that of ice[51]. A systematic analysis of hybrid SCAN functionals with varying fractions of Hartree−Fock exchange demonstrated the inability of these functionals to accurately represent 2-body interactions between water molecules, with errors up to ~5 kcal/mol for the water hexamer relative to reference values calculated using coupled cluster theory with single, double, and perturbative triple excitations, i.e., CCSD(T), in the complete basis set (CBS) limit[51], the "gold standard" method for molecular interactions[53]. In this context, a neural-network potential, NNP-SCAN0, was recently trained on a modified SCAN0 functional that incorporates 10% Hartree−Fock exchange[52]. (It is worth noting that, in its original formulation, the SCAN0 functional mixes 25% Hartree−Fock exchange with 75% SCAN exchange[54].) Despite providing better agreement with experimental data than SCAN for several properties of liquid water measured at ambient conditions, this improved agreement was achieved by actually performing the NNP-SCAN0 simulations at 330 K[52].

While all previous studies suggest that SCAN is overall one of the most accurate XC functionals, they also indicate that any further improvement of the accuracy of DFT models for water requires removing, at least partially, the associated density-driven errors. To this end, we introduce here a data-driven many-body potential energy function (PEF) for water, MB-SCAN(DC), which is rigorously derived within a many-body formalism applied to density-corrected SCAN (DC-SCAN) data for individual many-body contributions to the interaction energies between water molecules. Density-corrected DFT (DC-DFT)[55–63], where the Hartree−Fock density is used instead of the Kohn-Sham density, is known to mitigate density-driven errors in GGA and meta-GGA functionals, especially nonempirical ones. In this regard, density-driven errors associated with calculations carried out on water clusters using the GGA PBE functional were found to be significant[63]. Here, we show that both binding and interaction energies calculated with the DC-SCAN functional for various water clusters are close to the CCSD(T)/CBS reference values, with DC-SCAN correctly reproducing each individual many-body contribution to the interaction energies. Importantly, we demonstrate that the MB-SCAN(DC) PEF preserves the accuracy of DC-SCAN and enables simulations of liquid water with significantly higher accuracy than all previous DFT-based models reported in the literature (including both ab initio and neural-network models), predicting structural, thermodynamic, and dynamical properties in quantitative agreement with experiment.

## Results
**Theoretical background.** In ground-state Kohn−Sham DFT[19], the energy is self-consistently minimized as:

$$E = \min_n \left\{ F[n] + \int d^3r \, n(\mathbf{r}) v(\mathbf{r}) \right\} \qquad (1)$$

where the minimizing $n(\mathbf{r})$ is the ground-state density, $v(\mathbf{r})$ is the external potential, and $F[n]$ includes the exact non-interacting kinetic and Hartree electrostatic energy terms plus an exchange-

correlation (XC) energy. Since the exact XC functional is unknown, different DFT approximations have been developed to solve Eq. (1). The total-energy error $\Delta E$ associated with different DFT approximations can be written as the sum of the functional-driven error, $\Delta E_F$, and the density-driven error, $\Delta E_D$[59]:

$$\Delta E = \Delta E_F + \Delta E_D \qquad (2)$$

The functional-driven error $\Delta E_F = E_{XC}^{approx}[n_{exact}] - E_{XC}^{exact}[n_{exact}]$ arises from the difference between the approximate XC functional, $F[n]$, and the (unknown) exact functional, while the density-driven error $\Delta E_D = E_{XC}^{approx}[n_{approx}] - E_{XC}^{approx}[n_{exact}]$ arises from using an approximate density $n(\mathbf{r})$ to solve Eq. (1). In most systems, the functional-driven error is the main contribution to the total error[56,59]. By many measures, the best nonempirical functionals predict more accurate densities for neutral atoms than the heavily parameterized empirical functionals or even Hartree−Fock theory[64]. But they still make density-driven delocalization errors[65,66] that can dominate the total error under special conditions[57,67].

Independent of the specific form and parametrization, standard approximate XC functionals still deviate from the piecewise-linear behavior of the exact functional for fractional charges[65], causing excess charge delocalization and resulting in incorrect densities[65,66]. For certain systems, the density-driven error thus become the dominant contributor to the total error[59,68]. This error can be understood by considering that the classical electrostatic repulsion term that is part of $F[n]$ in Eq. (1) contains a self-interaction contribution due to each electron interacting with itself[45,67]. While this self-interaction contribution should, in theory, be compensated by the XC energy, approximate XC functionals contain substantial local components that prevent them from quantitatively removing electron self-interactions. As a result, the electron density thus tends to over-delocalize in order to minimize the many-electron self-interactions[48,69,70], leading to fractional charges that underestimate the energy predicted by the piecewise-linear behavior of the exact functional[65,71].

Using a more accurate density can mitigate errors due to the over-delocalization of the electron density[56,60,72]. However, obtaining an accurate density from wavefunction theories, such as Møller−Plesset peturbation theory and coupled-cluster theory, is computationally significantly more expensive than the corresponding DFT calculations. An approximate, yet efficient, approach to reducing density-driven errors in DFT calculations consists in using the Hartree−Fock density, $n^{HF}(\mathbf{r})$ because, by construction, it does not suffer from either electron over-delocalization or self-interaction errors[56,59–61,63]. The resulting density-corrected DFT (DC-DFT) energy can then be written as:

$$E^{DC-DFT} \approx E^{HF} + \left( E_{XC}^{approx}[n^{HF}] - E_X^{HF} \right) \qquad (3)$$

The occupied Hartree−Fock (HF) orbitals are used here in place of those from a self-consistent calculation with the approximated functional. If $E_{xc}^{approx}$ is not fitted to any bonded system, then neither is Eq. (3). Equation (3) takes advantage of the understood overall cancellation between semilocal approximations to the exchange and correlation energies; there is no corresponding overall cancellation in the potentials or functional derivatives. In meta-GGA functionals, the exchange-correlation energy depends explicitly not only on the electron density but also on the non-interacting kinetic energy density, and both these ingredients differ from HF to Kohn−Sham (KS) theory, but, importantly, for a meta-GGA functional like SCAN both the HF and the KS kinetic energy densities can be used to recognize iso-orbital and uniform density limits and to interpolate between them. In extensive molecular tests, SCAN evaluated on the Hartree−Fock density was found on average to be more accurate than self-

consistent SCAN, and even more accurate than all but a few hybrid functionals[73]. It is very possible for a density functional to yield accurate energies on physical densities and yet have an inaccurate functional derivative and thus an inaccurate self-consistent electron density, because the functional derivative yields the response of the functional to an arbitrary (and not necessarily physical) small density variation. For example, a self-consistent semilocal functional like SCAN cannot bind a full extra electron to an isolated neutral atom, but using the Hartree−Fock density for the negative ion yields an accurate electron affinity from such functionals[74]. It should be noted that density correction also has some limitations: (1) It can only correct part of the error of an approximate density functional. (2) Because it is not self-consistent, it cannot provide Hellmann−Feynman forces on the nuclei. (3) Going beyond the level of the Hartree−Fock approximation can incur not only the cost of the higher-level density but also the cost of inverting it to find an effective one-electron potential[75].

Equation (3) can be used to calculate individual $n$-body energies, $\epsilon^{nB}$, from one-body (1B) to $N$-body (NB), which enter the many-body expansion (MBE) of the energy for a system containing $N$ (atomic or molecular) monomers[76]:

$$E_N(1, \dots, N) = \sum_{i=1}^{N} \epsilon^{1B}(i) + \sum_{i<j}^{N} \epsilon^{2B}(i,j)$$
$$+ \sum_{i<j<k}^{N} \epsilon^{3B}(i,j,k) + \cdots + \epsilon^{NB}(1, \dots, N), \qquad (4)$$

In the case of water, $\epsilon^{1B}(i)$ in Eq. (4) corresponds to the distortion energy of the $i$th water molecule in the system from the equilibrium geometry of the corresponding free molecule, and all higher-order $n$-body energies, $\epsilon^{nB}(2,\dots,n)$, can be calculated recursively from the lower-order terms[77]. The data-driven many-body formalism originally introduced with the MB-pol[78–81] potential energy function (PEF) for water proceeds from monomer to dimer to trimer, successively calculating the 1B, 2B, and 3B energies over a wide distribution of molecular configurations and then fitting them to analytic potential energy functions. The 4B and higher-order terms of Eq. (4) are replaced by a classical many-body polarization model. Once the PEFs are known, fast but very accurate classical molecular dynamics calculations for finite temperature or fully relaxed geometry optimizations for zero temperature can be performed. It should be noted that for the equilibrium geometries of molecules and solids self-consistent SCAN already performs quite well[41,42]. Here, we show that DC-SCAN provides an accurate but inexpensive alternative to the accurate but expensive CCSD(T) parametrization of a data-driven many-body PEF for water, suggesting many possible future applications of DC-SCAN that we are beginning to explore.

**2-body interactions in water**. Our analysis of the ability of the SCAN functional to represent the interactions between water molecules begins with the comparison in Fig. 1 between the total 2-body (2B) energies (second term on the right-hand side of Eq. (4)) calculated for the low-energy isomers of the water hex-amer using the (self-consistent) SCAN and SCAN0 functionals, and the corresponding (density-corrected) DC-SCAN and DC-SCAN0 functionals. Also shown for reference are the CCSD(T)/CBS values reported in ref. [78]. It should be noted that the hexamer holds a special space along the path that connects individual water molecules in the gas phase to liquid water since it is the smallest water cluster for which the low-energy isomers are characterized by three-dimensional arrangements that are remi-niscent of the three-dimensional structure of the hydrogen-bond network found in the liquid phase. In addition, the large number

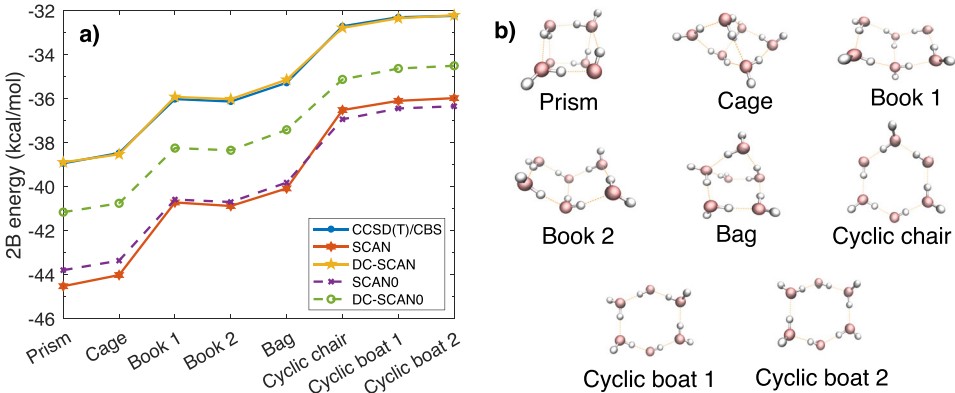

**Fig. 1 2B energies of the water hexamer isomers.** Two-body energies calculated for the first eight low-energy isomers of the water hexamer using SCAN, DC-SCAN, SCAN0 (with 25% exact exchange), and DC-SCAN0, along with the corresponding CCSD(T)/CBS reference values from ref. [78]. The 2-body energy, on average, contributes ~80−85% to the total interaction energy in water[14]. The errors associated with a given functional relative to the CCSD(T)/CBS values are roughly the same for each isomer.

of low-energy isomers makes the hexamer cluster the prototypical system to assess the ability of different water models to correctly reproduce many-body interactions in water[14]. Figure 1 shows that the SCAN functional displays fairly large errors compared to the reference values, with a maximum unsigned error (MUE) of 4.59 kcal/mol. In contrast, DC-SCAN predicts 2-body energies that are in quantitative agreement with the CCSD(T)/CBS values, resulting in a MUE of only 0.08 kcal/mol. By effectively eliminating the errors in the representation of 2-body interactions, the application of the density correction thus addresses the main shortcoming of the SCAN functional applied to water[50]. Figure 1 also shows that SCAN0, the hybrid variant of SCAN with a 25% fraction of Hartree−Fock exchange, only provides a minor improvement in the representation of the 2-body energies, resulting in a MUE of 4.48 kcal/mol. It should be noted that SCAN0 provides a slightly more accurate description of the three-dimensional isomers (i.e., prism and cage isomers) but a worse description of the planar isomers (i.e., cyclic isomers) compared to SCAN. Importantly, the density correction applied to SCAN0 does not result in a similarly dramatic improvement as found for SCAN, with DC-SCAN0 still displaying a relatively large MUE of 2.26 kcal/mol.

The analysis of the effects associated with various dispersion corrections, which is reported in Supplementary Fig. 1, indicates that the addition of any form of dispersion energy worsens the accuracy of both SCAN and DC-SCAN. Specifically, all the dispersion-corrected SCAN functionals considered in our analysis, SCAN-D3(0), SCAN-D3(BJ), and SCAN-VV10, are found to overbind the hexamer isomers, which results in larger deviations from the CCSD(T)/CBS values compared to their dispersion-free counterparts. Similarly, inclusion of larger fractions ($\alpha$) of Hartree−Fock exchange deteriorates the ability of hybrid SCAN$\alpha$ and DC-SCAN$\alpha$ functionals to reproduce the interaction energies of the hexamer isomers, resulting in significant overbinding (Supplementary Fig. 2). It is also worth noting that neither DC-SCAN + dispersion nor DC-SCAN$\alpha$ (where $\alpha$ is the fraction of HF exchange) perform as well as DC-SCAN. This suggests that the addition of the dispersion correction and/or a fraction of Hartree−Fock exchange actually worsens the functional-driven error of SCAN for water.

To further investigate the impact of the density correction on the energetics of various water systems, in Fig. 2, we analyze the interaction energies of dimers extracted from a classical MD simulation of liquid water carried out in the isobaric-isothermal (NPT) ensemble at ambient conditions using the MB-pol PEF[79−81]. (The interaction energy is the binding energy without its 1-body contribution.) For this analysis, we consider dimers

with an oxygen−oxygen (O ⋯ O) distance shorter than 5.5 Å, which approximately corresponds to the radius of the first two solvation shells in liquid water[81−83]. It should be noted that, by definition, the interaction energy of a water dimer exactly corresponds to the associated 2-body energy. Figure 2a shows the errors, $\Delta E$, in 2-body energies calculated with SCAN and DC-SCAN relative to the corresponding reference values calculated at the CCSD(T)-F12b level of theory. As expected, DC-SCAN exhibits significantly smaller errors compared to SCAN for all dimers, independent of the O ⋯ O distance. Specifically, the maximum error associated with DC-SCAN is −0.47 kcal/mol, which must be compared with a maximum error of −1.38 kcal/mol calculated with SCAN. It is also important to analyze the errors as a function of the O ⋯ O distance since they directly affect the ability of the SCAN and DC-SCAN functionals to correctly predict the cohesive energy, and thus the structure, of liquid water. Figure 2a shows that the 2-body energies calculated with SCAN only start to approach the CCSD(T)-F12b values at ~4.5 Å, with a MUE of 0.25 kcal/mol associated with dimers with an O ⋯ O distance up to 4.5 Å, and a MUE of 0.16 kcal/mol for all dimers up to an O ⋯ O distance of 5.5 Å. In contrast, the 2-body energies calculated with DC-SCAN converge to the CCSD(T)-F12b values at 3.5 Å, with a MUE of 0.09 kcal/mol obtained for dimers with an O ⋯ O distance up to 3.5 Å, which decreases to 0.07 kcal/mol when all dimers up to an O ⋯ O distance of 5.5 Å are considered.

Figure 2b shows a comparison between the interaction energies calculated at the CCSD(T)-F12b, SCAN, and DC-SCAN levels of theory for an unrelaxed scan of the water dimer along the O ⋯ H distance, starting from the dimer optimized geometry. This comparison provides further evidence for DC-SCAN predicting 2-body energies in close agreement with the CCSD(T)-F12b values. In contrast, SCAN systematically overbinds the water dimer, which is particularly evident in the minimum-energy region, i.e., r(O ⋯ H) ~ 1.9 Å. It is worth noting that SCAN gives slightly better agreement with CCSD(T)-F12b at long range, before asymptotically converging to DC-SCAN results. The reason behind SCAN giving slightly better result than DC-SCAN at long range is the cancellation in SCAN of the density-driven errors (which overbind) with the lack of dispersion.

**Binding energies of water clusters**. It is known that the binding energies of the low-energy isomers of the water hexamer lie within a few kcal/mol from each other[84] while the two most stable isomers ($D_{2d}$ and $S_4$) of the water octamer are degenerate[85]. Table 1 shows that the SCAN functional predicts significantly

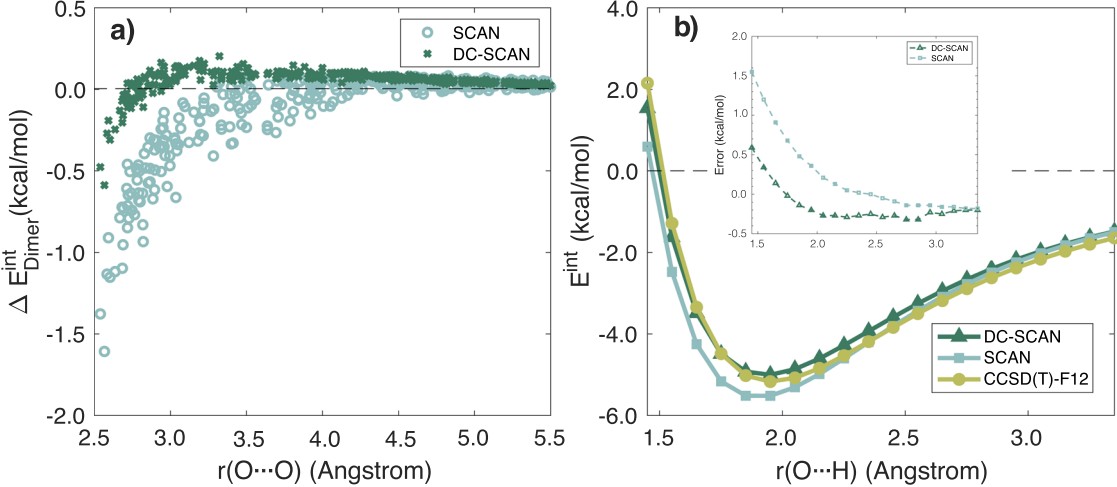

**Fig. 2 Comparison of dimer interaction energies. a** Errors in 2-body energies calculated with SCAN and DC-SCAN relative to CCSD(T)-F12b values for dimers with an oxygen−oxygen distance shorter than 5.5 Å which were extracted from an NPT simulation of liquid water carried out with MB-pol[79−81] at ambient conditions. **b** CCSD(T)-F12b, SCAN and DC-SCAN interaction energies calculated for an unrelaxed scan of the water dimer along the O ⋯ H distance. The inset of panel (**b**) shows the errors associated with DC-SCAN and SCAN relative to the CCSD(T)-F12b reference values as a function of r(O ⋯ H).

<table>
<tr><td colspan="4">

**Table 1 Errors (in kcal/mol) in binding energies relative to the CCSD(T)-F12b values of ref. [86] calculated for representative isomers of the water hexamer and octamer using SCAN, FLOSIC-SCAN (from ref. [50]) and DC-SCAN. The values in parentheses correspond to the errors per molecule. The last row reports the corresponding MUEs and MUEs per molecule.**

</td></tr>
</table>

| | SCAN | FLOSIC-SCAN | DC-SCAN |
|---|---|---|---|
| Hexamer: Prism | 5.69 (0.95) | −1.62 (−0.27) | −0.73 (−0.12) |
| Hexamer: Cage | 5.68 (0.95) | −1.66 (−0.28) | −0.62 (−0.10) |
| Hexamer: Book 2 | 5.42 (0.90) | −1.56 (−0.26) | −0.41 (−0.07) |
| Hexamer: Cyclic boat 2 | 4.79 (0.80) | −2.43 (−0.41) | −0.12 (−0.02) |
| Octamer: $D_{2d}$ | 8.84 (1.10) | −1.31 (−0.16) | −1.15 (−0.14) |
| Octamer: $S_4$ | 8.84 (1.10) | −1.31 (−0.16) | −1.13 (−0.14) |
| MUE | 6.54 (0.97) | 1.65 (0.26) | 0.69 (0.09) |

different binding energies relative to the CCSD(T)-F12b results of ref. [86] for both sets of clusters, with an overall MUE of 6.54 kcal/mol. Interestingly, the error per water molecule is higher for the three-dimensional isomers (prism and cage isomers) than for the planar isomers of the water hexamer, and increases for the two isoenergetic isomers of the octamer.

Table 1 also includes the binding energies calculated in ref. [50] with SCAN corrected for the self-interaction energy using the Fermi−Lowdin orbital self-interaction correction (FLOSIC) scheme[87]. Relative to SCAN, FLOSIC-SCAN is able to reduce the errors in the binding energies of all water clusters analyzed in Table 1, resulting in a MUE of 1.65 kcal/mol. The error per water molecule remains nearly constant for the prism, cage, and book-2 isomers of the water hexamer but increases for the cyclic boat-2 isomer. The FLOSIC-SCAN error per molecule is smaller for the two isoenergetic isomers of the water octamer. The comparisons reported in Table 1 show that DC-SCAN performs better than FLOSIC-SCAN, with an overall MUE of 0.69 kcal/mol relative to CCSD(T)-F12b. As found with FLOSIC-SCAN, also in the case of DC-SCAN the error per molecule remains constant for the prism, cage, and book-2 isomers of the water hexamer but decreases for the cyclic boat-2 isomer. However, contrary to FLOSIC-SCAN,

DC-SCAN predicts a slightly larger error per molecule for the two isoenergetic octamer isomers than for the prism, cage, and book-2 isomers of the hexamer. The overall MUE per molecule of 0.04 kcal/mol indicates that the binding energies predicted by DC-SCAN are in excellent agreement with the CCSD(T)-F12b reference values for all clusters analyzed in Table 1.

**Many-body interactions in water**. Although the results presented in Fig. 1 and Table 1 demonstrate that, by correcting density-driven errors, DC-SCAN is able to accurately reproduce the interaction energies of small water clusters, the analyses of the previous sections do not provide any direct information about the ability of DC-SCAN to correctly describe many-body effects in water. The competition and interplay of many-body effects have been shown to play a critical role in determining structural, thermodynamic, and dynamical properties of aqueous systems, from small clusters to bulk solutions and interfaces[84,88−91].

To investigate the impact of the density correction on individual $n$-body ($n$B) contributions to the interactions between water molecules, many-body decomposition analyses were carried out for the two isoenergetic isomers of the octamer. Errors relative to the CCSD(T)-F12b reference values are shown in Fig. 3 for each $n$-body energy calculated with the SCAN and DC-SCAN functionals. This analysis provides further evidence for the density-driven errors in the SCAN functional primarily affecting 2-body energies, with SCAN displaying large negative deviations from the CCSD(T)-F12b values, which confirms the tendency of the SCAN functional to overbind water clusters[50]. After application of the density correction, the errors in the 2-body energies reduce to only ~0.3 kcal/mol for calculations carried out with the DC-SCAN functional. Importantly, Fig. 3 shows that the impact of the density correction is minimal for all $n$B energies with $n > 2$.

After demonstrating that, by removing density-driven errors, the DC-SCAN functional effectively provides chemical accuracy for binding, interaction, and many-body energies of various water clusters, the 2B, 3B, and 4B energies, as well as the total interaction energies of the low-energy isomers of the water hexamer calculated using the SCAN and DC-SCAN functionals are compared in Fig. 4 with the analogous values calculated with the corresponding MB-SCAN and MB-SCAN(DC) potential

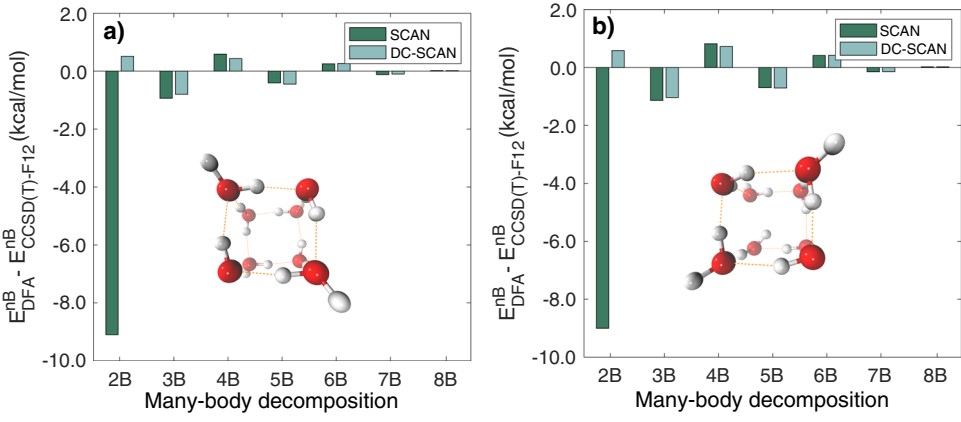

**Fig. 3 Errors in *n*B interaction energy of water octamers.** Errors relative to CCSD(T)-F12b reference values for each *n*B energy contribution to the interaction energies calculated for the two isoenergetic isomers, **a** $D_{2d}$ and **b** $S_4$, of the water octamer using the SCAN and DC-SCAN functionals.

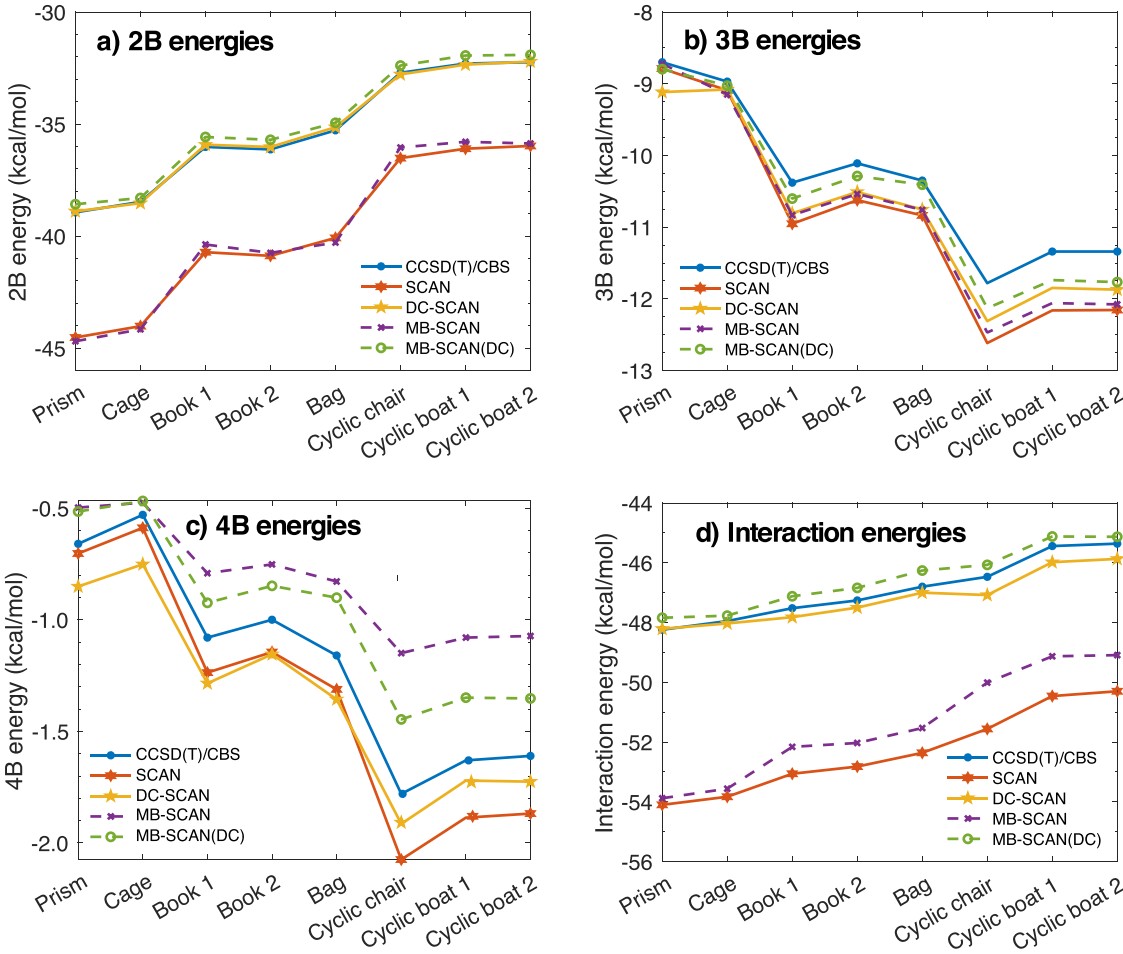

**Fig. 4 Many-body and interaction energies for the isomers of the water hexamer. a** 2-body (2B), **b** 3-body (3B), **c** 4-body (4B) and **d** total interaction energies of the first eight isomers of the water hexamer calculated using SCAN, DC-SCAN, MB-SCAN, MB-SCAN(DC), along with the CCSD(T)/CBS reference values of ref. [78].

energy functions (PEFs) described in the "Methods" section. Also shown for reference are the CCSD(T)/CBS values reported in ref. [78]. The errors relative to the reference CCSD(T)/CBS values for both many-body and interaction energies which are associated with the SCAN and DC-SCAN functionals and the corresponding MB-SCAN and MB-SCAN(DC) PEFs are shown in Supplementary Fig. 7. As already discussed in the case of the octamer isomers, density-driven errors are most pronounced at the 2B

level, with MUEs of 4.59 and 0.08 kcal/mol associated with SCAN and DC-SCAN, respectively. The MUEs reduce to 0.59 and 0.38 kcal/mol at the 3-body level. The comparisons shown in Fig. 4a, b demonstrate that both the MB-SCAN and MB-SCAN(DC) PEFs are able to quantitatively reproduce the 2-body and 3-body energies calculated ab initio with the corresponding SCAN and DC-SCAN functionals. Since, by construction, *n*B energies with *n* > 3 in the MB-SCAN and MB-SCAN(DC) PEFs

are entirely represented by a classical polarization term, the errors associated with these energies are not strictly related to those calculated ab initio with the corresponding SCAN and DC-SCAN functionals. In this regard, Fig. 4c shows that the 4-body energies predicted by the MB-SCAN and MB-SCAN(DC) PEFs tend to underbind the hexamer isomers relative to CCSD(T)/CBS, whereas the 4-body energies calculated with the SCAN and DC-SCAN functionals tend to overbind the same clusters. However, it should be noted that in both cases the 4-body errors are small for all eight isomers, with SCAN and MB-SCAN providing MUEs of 0.17 and 0.35 kcal/mol, respectively. The corresponding MUEs for DC-SCAN and MB-SCAN(DC) are 0.16 and 0.21 kcal/mol, respectively.

The total interaction energies of the eight low-energy hexamer isomers calculated with the SCAN and DC-SCAN functionals, and the corresponding MB-SCAN and MB-SCAN(DC) PEFs are compared with the CCSD(T)/CBS reference values in Fig. 4d. Both DC-SCAN and MB-SCAN(DC) provide excellent agreement with the CCSD(T)/CBS reference values, displaying MUEs of 0.53 and 0.36 kcal/mol, respectively. In contrast, suffering from large density-driven errors at the 2-body level, SCAN and MB-SCAN systematically overbind all eight isomers.

**Structural and dynamical properties of liquid water.** The last question that remains to be addressed is whether the high accuracy displayed by the MB-SCAN(DC) PEF in reproducing the multidimensional energy landscape of water clusters is sufficient to correctly predict the properties of liquid water. To this end, classical MD simulations for a periodic box containing 256 molecules were carried out with the MB-SCAN(DC) PEF in the NPT ensemble at 1 atm and various temperatures between $T = 250$ K and $T = 350$ K. The lengths of the MD trajectories were 2.6 ns for $T < 298$ K and 2 ns for $T \geq 298$ K. Figure 5 shows that the MB-SCAN(DC) PEF correctly reproduces the temperature-dependence of the density of liquid water at 1 atm, underestimating the experimental values by only ~0.01 g/cm³ at all temperatures. At 298 K, MB-SCAN(DC) predicts a density of 0.986 g/cm³, which is in close agreement with the experimental value of 0.997 g/cm³. The temperature of maximum density calculated by fitting a fifth-order polynomial to the MB-SCAN(DC) results is 280 K, in nearly quantitative agreement with the

experimental value of 277 K. The MB-SCAN(DC) results are compared in Fig. 5 with those reported in the literature from MD simulations with SCAN[43] (SCAN-AIMD) as well as with NNPs trained on SCAN[92] (SCAN-NNP) and SCAN0[52] (SCAN0-NNP) data. These comparisons demonstrate that the MB-SCAN(DC) PEF predicts a liquid density at 330 K which is in significantly closer agreement with experiment than the value calculated in ref. [43] from ab initio MD simulations with SCAN.

Particularly interesting is the comparison of the MB-SCAN(DC) PEF with the two NNPs models trained on SCAN[92] and SCAN0[52] data. Figure 5 shows that, despite being trained on SCAN data, the SCAN-NNP model is unable to correctly reproduce the density value calculated from ab initio MD simulations with SCAN at 330 K. A closer agreement between the density values calculated from the SCAN-AIMD and SCAN-NNP simulations was obtained after applying a reweighting procedure[92]. In addition, the SCAN-NNP model predicts a more pronounced temperature-dependence of the liquid density compared to experiment, overestimating both the value and the temperature of the density maximum[92]. A slightly more accurate prediction of the liquid density at 330 K is provided by the SCAN0-NNP model[52], although no ab initio MD simulations with SCAN0 have been reported to compare with. Given the increased popularity of NNPs trained on DFT data, we believe that the differences between SCAN-AIMD and SCAN-NNP results deserve further investigation to assess the ability of NNPs to faithfully represent the target DFT models. In this context, it should be noted that in a previous study[51] we found that MD simulations carried out with the MB-SCAN PEF predict a liquid density of 1.14 g/cm³ at 298 K, which is significantly different from the value of 1.05 g/cm³ obtained from ab initio simulations with SCAN[43]. This difference is not due to the different size of the water systems studied in the two sets of simulations (256 molecules for MB-SCAN[51] and 64 molecules for SCAN-AIMD[43].) An explanation for this difference, proposed in ref. [51], considers that any PEF rigorously derived from the many-body expansion of the energy (MBE) is strictly faithful to its parent quantum-mechanical method only when the latter does not display spurious delocalization of the electron density which affects the convergence of the MBE in an unphysical manner. We believe that the present analysis of the SCAN and DC-SCAN functionals, along with the corresponding MB-SCAN and MB-SCAN(DC) PEFs, provides support for the interpretation presented in ref. [51] that density-driven errors are responsible for the differences between MD simulations carried out with the SCAN functional and the MB-SCAN PEF. The temperature-dependence of the enthalpy of vaporization and isothermal compressibility calculated from classical MD simulations with MB-SCAN(DC) are shown in Supplementary Figs. 11 and 12.

Figure 6 compares the oxygen−oxygen ($g_{OO}$) radial distribution function (RDF) calculated from MD simulations carried out with the MB-SCAN and MB-SCAN(DC) PEFs at 298 K and 1 atm with the corresponding experimental data[82,83]. The MB-SCAN(DC) PEF provides excellent agreement with the experimental RDF, slightly overestimating the height of the first peak while underestimating the height of the "valley" between the first two peaks. As shown in Supplementary Fig. 10, these small differences can be attributed to the neglect of nuclear quantum effects in classical MD simulations. The inclusion of nuclear quantum effects in path-integral molecular dynamics (PIMD) simulations with MB-SCAN(DC) indeed slightly lowers the height of the first peak and raises the "valley" between 3.2 and 4.0 Å similarly to what previously observed in the $g_{OO}$ calculated with the MB-pol PEF[81]. As expected, the inclusion of nuclear quantum effects also improves the agreement with the experimental oxygen−hydrogen and hydrogen−hydrogen RDFs

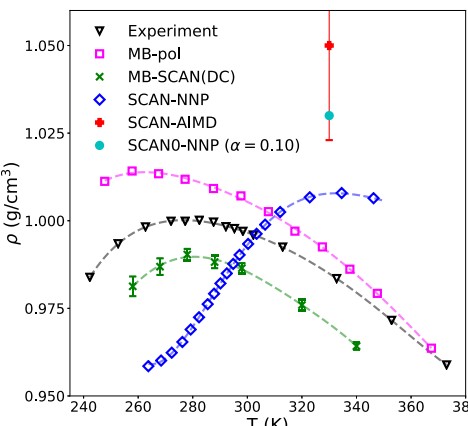

**Fig. 5 Density of liquid water.** Temperature-dependence of the density of liquid water at 1 atm calculated from classical NPT simulations carried out with MB-SCAN(DC) along with the results from SCAN-AIMD[43], SCAN-NNP[92], and SCAN0-NNP (with 10% HF exchange)[52] simulations. The MB-pol results are from ref. [78], while the experimental data are from the NIST Chemistry WebBook[124]. Error bars for the MB-SCAN(DC) results represent 95% confidence intervals.

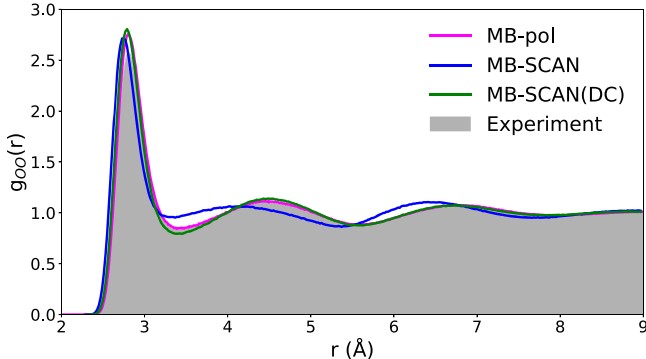

**Fig. 6 Structure of liquid water.** Oxygen−oxygen ($g_{OO}$) radial distribution function (RDF) calculated from NPT simulations carried out with the MB-SCAN(DC) PEF at 298 K and 1 atm. The MB-pol RDF is from ref. [81], while the experimental RDF at 295 K is from ref. [82].

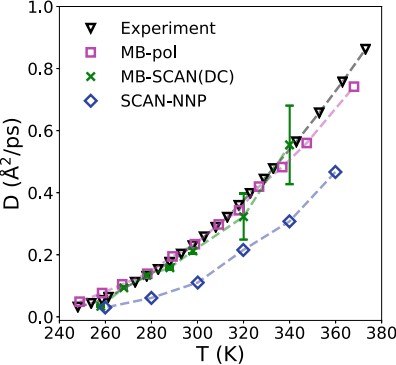

**Fig. 7 Self-diffusion of liquid water.** Temperature-dependence of the self-diffusion coefficient of liquid water calculated from NVE simulations carried out with the MB-SCAN(DC) PEF. The SCAN-NNP data are from ref. [93], the MB-pol results are from ref. [78], while the experimental data are from refs. [125,126,127]. Error bars for the MB-SCAN(DC) results represent 95% confidence intervals.

(Supplementary Fig. 10). In contrast, as already discussed in ref. [51], the MB-SCAN PEF predicts a denser and more unstructured liquid. Based on the analyses discussed above, the differences between the MB-SCAN and MB-SCAN(DC) oxygen−oxygen RDFs can be unambiguously attributed to density-driven errors that affect SCAN many-body energies, particularly at the 2-body level, which are used to train the corresponding MB-SCAN PEF.

To provide further insights into the ability of the MB-SCAN(DC) PEF to describe the properties of liquid water, we also calculated the temperature-dependence of the self-diffusion coefficient, $D$, from a 500-ps-long MD simulation carried out in the microcanonical (NVE) ensemble for a periodic box containing 256 molecules using the equilibrium density determined from the corresponding NPT simulations. $D$ was calculated from the velocity autocorrelation function of the center of mass of each water molecule according to

$$D = \frac{1}{3}\int_0^\infty \langle v_i(t)v_i(0)\rangle \mathrm{d}t, \tag{5}$$

where $v_i$ is the center of mass velocity of the $i$th water molecule. Figure 7 shows that the MB-SCAN(DC) PEF is able to correctly predict the diffusion coefficient between 250 and 350 K. In particular, at 298 K, the diffusion coefficient predicted by the MB-SCAN(DC) PEF is 0.212 A$^2$/ps, which is in excellent agreement with the experimental value of 0.229 A$^2$/ps. It should be noted that the larger error bars associated with the values of the diffusion coefficient at higher temperatures are due to larger fluctuations in the molecular velocities. This is in contrast to the value of 0.106 A$^2$/ps obtained in ref. [93] from MD simulations carried out with an adaptive neural-network model trained on SCAN data (SCAN-NNP in Fig. 7). In contrast to the MB-SCAN(DC) PEF, the SCAN-NNP model severely underestimates the diffusion coefficient of liquid water over the entire temperature range, although the agreement with experiment apparently improves as the temperature decreases.

### Discussion

An ab initio representation of water across all the different phases has been an elusive goal since the early days of computer simulations[94–97]. Although models based on correlated wave-function theories (WFT) can, in principle, provide such a long-sought after ab initio representation of water without resorting to ad hoc approximations or empirical parameterizations, the associated computational cost precludes the application of WFT models to systems containing more than a handful of water

molecules. This effectively leaves DFT as the only viable approach to ab initio simulations of water[13]. However, it has been shown that existing XC functionals are not particularly accurate in their predictions of the properties of water[14,37], suffering from both functional-driven and density-driven errors.

In this study, we have demonstrated that the density-corrected SCAN (DC-SCAN) functional effectively removes density-driven errors from the water 2-body energies, which brings both binding and interaction energies of different water clusters very close to reference values calculated at CCSD(T)/CBS level of theory. Although not as pronounced as for the 2-body energies, the density correction also reduces density-driven errors in all higher-body terms of the many-body expansion (MBE) of the energy calculated for water using the DC-SCAN functional, with each individual many-body term being in quantitative agreement with the corresponding CCSD(T)/CBS reference values. In this context, it should be noted that a previous study[50] found a significant but less complete improvement for water clusters (Table 1) via a self-consistent FLOSIC self-interaction correction to SCAN. However, ref. [50] did not find evidence for a major improvement from density correction, probably because the FLOSIC density is less localized than the Hartree−Fock and exact densities are. Although it should be kept in mind that the DC-SCAN functional, as does the parent SCAN functional, still suffers from functional-driven errors, which can be large for some chemical systems such as stretched H$_2^+$, the analyses presented here demonstrate that these functional-driven errors are negligible for water. In the future, it would be important to test the performance of DC-SCAN for more-general chemical applications. Importantly, our analyses suggest that, in principle, ab initio MD simulations with the DC-SCAN functional should be able to provide a consistently accurate description of the properties of water. However, the requirement of using the Hartree−Fock density in a non-self-consistent SCAN calculation at each MD step would make ab initio MD simulations with DC-SCAN not straightforward to implement and expensive to perform.

While ab initio MD simulations with DC-SCAN are currently not feasible, we have shown that the improved accuracy of the DC-SCAN functional can be exploited to develop a data-driven many-body potential energy function, the MB-SCAN(DC) PEF, which indeed provides a highly accurate representation of water, from small clusters in the gas phase to the liquid phase. MB-SCAN(DC) is rigorously derived from the DC-SCAN MBE and adopts a hybrid data-driven/physics-based scheme, where a data-driven model, which captures (short-range) quantum-mechanical

interactions arising from the overlap of the electron densities of individual molecules at the 2-body and 3-body levels (e.g., Pauli repulsion, and charge transfer and penetration), is integrated with a physics-based model of many-body interactions, which is represented by classical many-body electrostatics. Importantly, we have demonstrated that the MB-SCAN(DC) PEF achieves high accuracy by quantitatively reproducing each individual term of the DC-SCAN MBE for water, providing a correct representation of both short- and long-range many-body contributions. Since the DC-SCAN functional exhibits chemical accuracy for each individual term of the MBE for water and the MB-SCAN(DC) PEF quantitatively reproduces the DC-SCAN many-body energies, the MB-SCAN(DC) PEF effectively provides the first demonstration of a DFT-based model that correctly describes the properties of water, at the computational cost of advanced polarizable force fields[14]. Future applications of the MB-SCAN(DC) PEF will focus on modeling the phase diagram of water, which was shown to be only qualitatively reproduced by NNPs trained on SCAN data[92,98]. We expect MB-SCAN(DC) to be especially well suited to modeling the liquid/vapor equilibrium, which involves the making and breaking of hydrogen bonds.

Finally, we want to emphasize that the many-body formalism adopted by the MB-SCAN(DC) PEF for water is general and has already been used in the development of data-driven many-body PEFs for various aqueous systems[99,100] and molecular fluids[101,102] which were trained on (expensive) CCSD(T) data. It thus follows that the significantly lower computational cost associated with DC-SCAN calculations can enable the routine development of MB-SCAN(DC) PEFs for generic (small) molecules which are trained on DC-SCAN data but effectively display CCSD(T) accuracy. In this context, it should be noted that the MB-Fit software infrastructure[103] for many-body PEFs combined with the MBX many-body energy/force calculator[104] interfaced with i-PI[105] and LAMMPS[106] already provides a robust platform for MD simulations of generic molecules in the gas, liquid, and solid phases using MB-SCAN(DC) PEFs.

## Methods

**Many-body expansion**. Building upon the demonstrated accuracy of the MB-pol PEF for water[78–81] and following the same theoretical/computational approach employed in the development of DFT-based many-body PEFs[51,90,107], we used Eq. (4) to develop a data-driven many-body PEF, MB-SCAN(DC), that consistently reproduces each term of the MBE for water calculated using the DC-SCAN functional. Briefly, MB-SCAN(DC) includes explicit representations of 1B, 2B, and 3B energies, and describes all higher-order $nB$ energy terms ($n > 3$) through classical many-body polarization. Specifically, $\epsilon^{1B}$ in Eq. (4) is represented by the Partridge–Schwenke PEF[108], while $\epsilon^{2B}$ and $\epsilon^{3B}$ are represented by terms describing permanent electrostatics, dispersion energy, and induction, which are combined with short-range permutationally invariant polynomials (PIPs)[109] fitted to reproduce 2B and 3B energies calculated with DC-SCAN for the same training sets of water dimers and trimers used in the development of MB-pol[79,80]. A detailed description of the theoretical and computational framework adopted in the development of data-driven many-body PEFs for water can be found in the original references[51,79,80,90,107]. It should be noted that, since our many-body PEFs directly target the underlying molecular interactions, differences in the representation of the 1-body (1B) term of Eq. (5) have been found to be negligible for modeling the properties of liquid water[107] and the air/water interface[110].

All DFT calculations were performed with the aug-cc-pVQZ basis set[111,112] using Q-Chem[113] quantum chemistry packages. Since the SCAN functional is particularly sensitive to the real-space grid, all SCAN and DC-SCAN calculations are performed on the highly dense Euler–Maclaurin–Lebedev (99,590) grid[114,115] (58,410 points per atom). In this regard, the results of a sensitivity analysis reported in Supplementary Table S1 suggest that the SG2 grid[116] (~8000 points per atom) should also be sufficient to converge SCAN calculations. In case only smaller grids are available, we recommend to use r²SCAN[117], which often achieves an accuracy similar to SCAN. Single-point energy calculations using explicitly correlated coupled cluster, CCSD(T)-F12b, theory[118] were performed in the CBS limit by extrapolating[119,120] the energy values obtained with the cc-pVTZ-F12 and cc-pVQZ-F12 basis sets along with associated auxiliary and complementary auxiliary (CABS) basis sets[121,122] using the ORCA quantum chemistry package[123].

## Data availability

All data generated and analyzed for this study are publicly available in this repository on GitHub: https://github.com/paesanilab/Data_Repository/tree/main/MBSCANDC.

## Code availability

The MB-SCAN and MB-SCAN(DC) PEFs are available in MBX[104], and can be used in MD simulations with LAMMPS[106] and i-PI[105]. All computer codes used in the analysis presented in this study are available from the authors upon request.

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

## Acknowledgements

We thank Eunji Sim, Suhwan Song, and Kieron Burke for stimulating discussions. This research was supported by the U.S. Department of Energy, Office of Science, Office of Basic Energy Science, through grants no. DE-SC0019490 (F.P.) and no. DE-SC0018331 (J.P.P.). This research used resources of the National Energy Research Scientific Computing Center (NERSC), which is supported by the Office of Science of the U.S. Department of Energy under contract DE-AC02-05CH11231, the Extreme Science and Engineering Discovery Environment (XSEDE), which is supported by the National Science Foundation through grant no. ACI-1548562, and the Triton Shared Computing Cluster (TSCC) at the San Diego Supercomputer Center (SDSC).

## Author contributions

S.D., E.L., J.P.P., and F.P. analyzed the data and wrote the paper. S.D. and E.L. contributed equally to this work. F.P. designed and supervised the research.

## Competing interests

The authors declare no competing interests.
