## [Peer Review File · Nature Communications]

REVIEWER COMMENTS

Reviewer #1 (Remarks to the Author):

The authors present an evaluation of DC-scan for liquid water using a series of relevant benchmark systems.

Their results show that this correction to the SCAN functional is very good, particularly for water. Hence the results indicate that the most important errors of scan, in particular related to its performance in evaluating liquid water are mostly density driven errors. This using the original functional and density driven errors as proposed by K. Burke and collaborators in the relevant (and well cited) papers (refs 56, 59).

Then given that this is an energy correction that can only be done in practice without self consistency and hence, not satisfying Hellman Feynman theorem to obtain the forces as derivatives of the energy. -Note that although in reference (J. Chem. Theory Comput. 2019, 15, 12, 6636–6646) a path to this selfconsistency is proposed, it is still not practical.

This implies that the method is only useful to obtain total energy correctors.

The authors however use this to show where the origins of the SCAN errors are with respect to water.

then they use the MB-POL method developed by Paesani to fit a force field to a DC-scan PES for water. This MB-DC-Scan is then used in classical MD simulations to ascertain properties of liquid water, showing that indeed this is a very accurate PES.

Overall the manuscript is well written. But I actually found it very confusing that the actual theory of the paper is buried in the methods section. I think that the paper should incorporate the DC-scan algorithmic developments before the results section. Otherwise it feels very confusing.

In regards to this, something that would make this paper more impactful is if the authors could explain what no other paper in DC-DFT has done so far.

I presume that in DC scan the kinetic energy density is obtained using the HF wave functions.

It is not clear to me if the self consistency in DC scan is done with SCAN or with HF. I'm referring to eq 4.

The E^{HF} is the self consistency converged HF energy. E^{HF}_X is the exchange part of that energy.

However, in $E^{\text{approx}}_{\text{XC}}[n^{\text{HF}}]$ one needs to compute also the kinetic energy density.

My question is, do the authors use the HF wave functions for that or the scan wave functions for that?

This is a very relevant question, because the kinetic energy density differs between HF and KS calculation that produces the HF density.

The only correct way to do it would be to invert the KS-equations to find the correct 1-body potential. This is possible, but computationally cumbersome and expensive. The authors should discuss how this is done in their DC-scan derivation. In general when others use DC-DFT they do it in functionals that are explicitly only density dependent. But a meta-GGA also depends on this additional term and so far no clear explanation appears in the literature for this.

The final result, although significant for DC-scan, seems not to be that relevant, given that MB-pol itself is still above DC-scan in performance. What is the advantage or motivation to use DC-SCAN for simulating liquid water instead of MB-pol?

So, right now I'm not convinced that the manuscript is impactful enough for Nat Communications. Not in its current form. It is a very nice manuscript but possibly for a more technical journal.

The way they sell their findings is a little strange. Basically the interesting result is how good density-corrected SCAN is. I bet the only reason they collaborated with Paesani on this is because they cannot obtain forces for density-corrected calculations. So to run MD calculations, they actually had to fit a force field (MB-Pol) to the energy reference values obtained with DC-SCAN. That the MB-Pol part works well is of course no surprise as they have done that before many times on both CCSD(T) as well as DFT data. So I guess the main insight of the paper is basically the accuracy of DC-SCAN. DC-SCAN is only practical for energy calculations or if an auxiliary force-field is fitted to it. Overall I'd say the paper is interesting but only intermediate impact.

Reviewer #2 (Remarks to the Author):

This work appears to establish that the SCAN functional, evaluated on HF densities, and then fed into the MB-pol procedure, yields very accurate energetics for water hexamers and excellent properties of bulk water. If this holds up for other liquids, it would provide a route to generating MB-pol accuracy forcefields for many more complex liquids, at DFT cost. This is definitely an important result with the potential to have high impact, and worth publishing in nat comms. But first the authors must address the following issues.

1. The title claims 'chemical accuracy' and indeed, Fig 4(b) shows DC-SCAN errors less than 1 kcal/mol. But the difference between MB-DC-SCAN and DC-SCAN is about 1 kcal/mol, so are the authors guaranteeing the true result is always between these two?
2. Throughout the manuscript, there should be additional lines appearing in most plots, which are the MB-pol results. This would allow readers to judge the relative error compared to the best MB results. It would also be useful to have all the numbers in the plots tabulated in supp info.
3. The previous point is especially important in Figs 5 + 6 + 7. Also, how large are nuclear effects here? One should plot the MB-pol results in MD and PIMD, rather than simple comparison to experiment.
4. Fig 4b suggests that MB-DC-SCAN gets the boats 1 and 2 energies in the wrong order, but one needs the actual numbers to tell.
5. An inset plotting errors for Fig 4(b) would help. Here DC-SCAN is worse than SCAN beyond about 2.1 Å. Why is that not a problem? Also, HF-DFT is supposed to yield accurate energies for stretched heterodimers, but it looks like the correction is vanishing at large R. Please explain.
6. The only explanation I can see for the DC-SCAN0 result in Fig 1 is that the mixing of 25% exchange substantially worsens the energy error, and that 0% exchange is best. This is an important result (showing that SCAN0 is worse for water than SCAN, once evaluated on a good density). If the authors run 50% or even 100% exact exchange, they could confirm this.
7. At the bottom of page 25, the second last sentence contains 'nonempirical' twice, when it should probably be 'empirical' the second time. But even then, it is too broad, as wB97 and double hybrids yield more accurate densities.
8. It would be useful for the authors to include 1 or 2 other density functionals for comparison. For example, PBE should also improve with HF density, but presumably not as much as SCAN does. Also, how do these results compare with those of omegaB97M-V?

Reviewer #3 (Remarks to the Author):

The authors report a study of accuracy of density-corrected SCAN (DC-SCAN), a flavour of meta-GGA DFT, for the description of water.

They benchmark DC-SCAN against coupled-cluster energies for water dimers and small water clusters, and assess its accuracy for thermodynamic properties of liquid water such as its density and O—O radial distribution function by comparing classical molecular dynamics (MD) simulations to experimental data. In order to render these MD simulations affordable they fit a force-field of MBpol architecture to DC-SCAN reference data, which is then used as a surrogate for first-principles DC-SCAN calculations.

Although the study is limited to a particular flavour of DFT and its accuracy for a single system — water — the detailed, yet clear, instructive, and convincing discussion of accuracy of DC-SCAN for describing the energetics of water dimers and clusters warrants publication.

Nonetheless, there are areas in which the manuscript could be improved.

While the comparison of DC-SCAN to CCSD(T)-CBS for the dimers and clusters benefits from the ability to compare both levels of theory on an equal footing, since computing energies is tractable in both cases, the assessment of the accuracy of DC-SCAN for thermodynamic properties of water is more complicated for two main reasons:

(1) the computational cost associated with DC-SCAN forces the use of MB-SCAN(DC) as a surrogate in the MD simulations and simultaneously prevents direct assessment of how reproduction errors impact thermodynamic observables.

(2) since reference quantum chemistry data is even more unaffordable, experiment provides the best measure of the performance of the potential. However, comparison with experiment in principle requires rigorously accounting for thermal AND quantum fluctuations of the nuclei. Unfortunately, the authors only present results of classical MD simulations here.

Indeed, both deuteration experiments and theoretical works such as www.pnas.org/cgi/doi/10.1073/pnas.1815117116 suggest quantum effects to increase the density of water at standard conditions (by of the order of 1%), promising even better agreement with experiment for MB-SCAN(DC) path-integral MD simulations (which should be rather affordable given the nature of the potential).

Similarly deuteration experiments suggest that quantum effects lead to a slight de-structuring of the O—O RDF of water, which again promises improved agreement with experiment for MB-SCAN(DC) path-integral MD simulations, compared to classical simulations.

In my view, both points warrant further discussion in the manuscript.

A few additional points are worth mentioning:

(1) In my view the abstract overreaches slightly by claiming that the work “demonstrates that MB-SCAN(DC) is effectively the first DFT-based model that correctly describes water from the gas to the condensed phase”, given that the condensed phase includes the full phase-diagram with all different crystalline forms and their diverse anomalous behaviour, some of which is driven by quantum-mechanical effects.

(2) With respect to Fig.1, a comment regarding the systematic offset and the comparatively very small errors in relative energy would be appreciated.

(3) Fig.2 supposedly shows that “DC-SCAN exhibits significantly smaller errors compared to SCAN for all dimers, independent of the O ··· O distance”. While the size of the markers used in panel (a) render assessing this difficult, panel (b) suggests this to be untrue above around 4.3 Angstrom, although the errors are very small at that point.

(4) A brief outline of the many-body expansion underlying the construction of the MB-SCAN(DC) potential would be appreciated.

(5) It remains unclear what configurations (and associated energies) underlie the MB-SCAN(DC) potential and why these could not have been computed using the gold standard, i.e. CCSD(T)-CBS, directly, given that CCSD(T)-CBS calculations for dimers and small clusters are clearly affordable.

On this note, the authors might want to highlight that a key benefit of the MB-SCAN(DC) potential over a hypothetical MB-CCSD(T)-CBS potential is that it provides evidence of accuracy of DC-SCAN, which in turn is transferable and hopefully accurate for other systems, for which CCSD(T)-CBS calculations are not viable.

(6) If the authors have investigated the errors in the thermodynamic observables (with respect to the first-principles reference) arising from the MB form/fitting, this would be an interesting addition to the manuscript. In particular, in view of the hypothesis that the reduced density errors in the DC-SCAN reference data facilitate the fitting.

Finally, the first half of the discussion appears to effectively reiterate the introduction. Personally, I feel that this repetition is redundant.

Response to Reviewer 1

The authors present an evaluation of DC-scan for liquid water using a series of relevant benchmark systems. Their results show that this correction to the SCAN functional is very good, particularly for water. Hence the results indicate that the most important errors of scan, in particular related to its performance in evaluating liquid water are mostly density driven errors. This using the original functional and density driven errors as proposed by K. Burke and collaborators in the relevant (and well cited) papers (refs 56, 59). Then given that this is an energy correction that can only be done in practice without self consistency and hence, not satisfying Hellman Feynman theorem to obtain the forces as derivatives of the energy. -Note that although in reference (J. Chem. Theory Comput. 2019, 15, 12, 6636–6646) a path to this selfconsistency is proposed, it is still not practical. This implies that the method is only useful to obtain total energy correctons. The authors however use this to show where the origins of the SCAN erros are with respect to water. then they use the MB-POL method developed by Paesani to fit a force field to a DC-scan PES for water. This MB-DC-Scan is then used in classical MD simulations to ascertain properties of liquid water, showing that indeed this is a very accurate PES.

We thank the Reviewer for careful reading of our manuscript, and comments, which are addressed below.

We have listed the limitations of density correction in a new paragraph before the paragraph of eq. 4: "It should be noted that density correction also has some limitations: (1) It can only correct part of the error of an approximate density functional. (2) Because it is not self-consistent, it cannot provide Hellmann-Feynman forces on the nuclei. (3) Going beyond the level of the Hartree-Fock approximation can incur not only the cost of the higher-level density but also the cost of inverting it to find an effective one-electron potential."

Overall the manuscript is well written. But I actually found it very confusing that the actual theory of the paper is buried in the methods section. I think that the paper should incorporate the DC-scan algorithmic developments before the results section. Otherwise it feels very confusing.

We slightly rearranged how we present both methods and results without modifying the format of Nature Communications. We hope this will help the reader better follow the flow of our manuscript.

In regards to this, something that would make this paper more impactful is if the authors could explain what no other paper in DC-DFT has done so far.

I presume that in DC scan the kinetic energy density is obtained using the HF wave functions. It is not clear to me if the self consistency in DC scan is done with SCAN or with HF. I'm referring to eq 4. The E^{HF} is the self consistency converged HF energy. E_X^{HF} is the exchange part of that energy. However, in $E_{XC}^{approx}[n^{HF}]$ one needs to compute also the kinetic energy density. My question is, do the authors use the HF wave functions for that or the scan wave functions for that?

This is a very relevant question, because the kinetic energy density differs between HF and KS calculation that produces the HF density. The only correct way to do it would be to invert the KS-equations to find the correct 1-body potential. This is possible, but computationally cumbersome and expensive. The authors should discuss how this is done in their DC-scan derivation. In general when others use DC-DFT they do it in functionals that are explicitly only density dependent. But a meta-GGA also depends on this additional term and so far no clear explanation appears in the literature for this.

It is true that for meta-GGA functionals the XC functional depends on the kinetic energy density as well as on the density and density gradient. The description of the non-interacting kinetic energy is different for the same density in Kohn-Sham and Hartree-Fock theories since the HF Slater determinant minimizes the Hamiltonian over all Slater determinants whereas the KS Slater determinant is restricted to orbitals derived from a single multiplicative potential. However, for a functional like SCAN, which satisfies all 17 known constraints for meta-GGA functionals, both the HF and the KS kinetic energy densities can be used to recognize iso-orbital and uniform density limits, and to interpolate between them.

Moreover it has been found that the inversion of the KS equations for the CCSD density shows a negligible improvement in dissociation curves over the HF density and the differences between the kinetic energies calculated using HF and KS theories are significantly smaller than the observed density driven errors (J. Chem. Theory Comput. 2020, 16, 5014–5023).

Following the Reviewer's suggestion, we have introduced a discussion about this point in the Introduction. In particular, below eq. 3, we have added: "The occupied Hartree-Fock (HF) orbitals are used here in place of those from a self-consistent calculation with the approximated functional. In meta-GGA functionals, the exchange-correlation energy depends explicitly not only on the electron density but also on the non-interacting kinetic energy density, and both these ingredients differ from HF to Kohn-Sham (KS) theory, but, importantly, for a meta-GGA functional like SCAN both the HF and the KS kinetic energy densities can be used to recognize iso-orbital and uniform density limits and to interpolate between them."

The final result, although significant for DC-SCAN, seems not to be that relevant, given that MB-pol itself is still above DC-scan in performance. What is the advantage or motivation to use DC-SCAN for simulating liquid water instead of MB-pol?

So, right now I'm not convinced that the manuscript is impactful enough for Nat Communications. Not in its current form. It is a very nice manuscript but possibly for a more technical journal.

The Reviewer is correct in noting that MB-pol for water already works very well. However, as we mention in the conclusions, we believe that the significantly lower computational cost associated with DC-SCAN calculations can enable the routine development of data-driven many-body MB-SCAN(DC) PEFs for generic molecules which are trained on DC-SCAN data but effectively display CCSD(T) accuracy. We believe that this is an important result, as also noted by Reviewer 2. In this regard, we are working on a follow-up study that identifies the classes of molecular systems for which DC-SCAN provides CCSD(T)-like accuracy for the underlying interactions, which will thus allow for using DC-SCAN in place of CCSD(T) for the development of data-driven MB-SCAN(DC) models transferable across different phases.

Our study also points out that density-driven errors may actually be the reason for the somewhat erratic behavior observed in *ab initio* MD simulations of liquid water with both SCAN (and other SCAN variants) and neural networks trained on SCAN data, with some properties being reproduced relatively well and other properties being appreciably different from the experimental data (e.g., Proc. Natl. Acad. Sci. USA 2017, 114, 10846; Proc. Natl. Acad. Sci. U.S.A. 2020, 117 26040; J. Chem. Theory Comput. 2021, 17, 3065; arXiv:2104.14410).

In addition, our study suggests that our many-body approach provides a data-driven approach distinct from neural networks which, consistently building upon a rigorous formulation of density-corrected energies, enables significantly more accurate simulations compared to neural networks trained on *ab initio* DFT-based simulations of liquid water. Given the widespread use of neural network potentials trained on DFT data, we believe that our study is particularly timely and important for the community interested in developing accurate data-driven models.

The way they sell their findings is a little strange. Basically the interesting result is how good density-corrected SCAN is. I bet the only reason they collaborated with Paesani on this is because they cannot obtain forces for density-corrected calculations. So to run MD calculations, they actually had to fit a force field (MB-Pol) to the energy reference values obtained with DC-SCAN. That the MB-Pol part works well is of course no surprise as they have done that before many times on both CCSD(T) as well as DFT data. So I guess the main insight of the paper is basically the accuracy of DC-SCAN. DC-SCAN is only practical for energy calculations or if an auxiliary force-field is fitted to it. Overall I'd say the paper is interesting but only intermediate impact.

This study was conceived and carried out by the Paesani group in collaboration with John Perdew, who had noticed appreciable self-interaction errors in the many-body analysis of the interaction energies in water clusters (Proc. Natl. Acad. Sci. USA 2020, 117, 11283). In this context, this study is the third of a series focusing on many-body interactions and density-driven errors in DFT representations of water. In the first paper (Ref. 51, J. Chem. Theory Comput. 2021, 17, 3739), we showed how errors in individual many-body energies calculated with SCAN α functionals (where α is the fraction of Hartree-Fock exchange) affect the ability of the corresponding MB-SCAN α models to describe liquid water. In the second paper (Ref. 92, now in press in the Journal of Chemical Theory and Computation, <https://doi.org/10.1021/acs.jctc.1c00541>), we demonstrated that exchange-correlation functionals commonly used in *ab initio* molecular dynamics simulations of liquid water are affected by significant density-driven errors, which stimulated the present study with DC-SCAN.

The Reviewer is correct in noting that our many-body formalism indeed provides an effective, yet accurate, way to map DC-SCAN into a MB-SCAN(DC) model that faithfully reproduces the DC-SCAN data and, at the same time, enables force calculations that are needed for MD simulations. As mentioned above, our MB-SCAN(DC) model thus allows for performing MD simulations at the DC-SCAN level of theory which are otherwise impossible to perform. We believe that this is an important result, especially when placed in context with state-of-the-art DFT-based neural networks that have so far struggled in properly modeling liquid water, being limited by the quality of DFT data available for bulk simulations.

Response to Reviewer 2

This work appears to establish that the SCAN functional, evaluated on HF densities, and then fed into the MB-pol procedure, yields very accurate energetics for water hexamers and excellent properties of bulk water. If this holds up for other liquids, it would provide a route to generating MB-pol accuracy forcefields for many more complex liquids, at DFT cost. This is definitely an important result with the potential to have high impact, and worth publishing in nat comms. But first the authors must address the following issues.

We thank the Reviewer for his/her/their careful reading and positive assessment of our manuscript. We address all comments below.

1. The title claims ‘chemical accuracy’ and indeed, Fig 4(b) shows DC-SCAN errors less than 1 kcal/mol. But the difference between MB-DC-SCAN and DC-SCAN is about 1 kcal/mol, so are the authors guaranteeing the true result is always between these two?

Based on our experience with MB-pol, we believe that DC-SCAN and the corresponding MB-SCAN(DC) PEF indeed achieve chemical accuracy for water. As shown in Fig. 4, both DC-SCAN and MB-SCAN(DC) are always within 1 kcal/mol from the CCSD(T) values for each individual many-body. Importantly, this translates in total interaction energies that are extremely close to the CCSD(T) reference data. While we cannot assess “chemical accuracy” for the liquid phase, we believe that the level of accuracy shown by MB-SCAN(DC) for small water clusters and various bulk properties suggests that MB-SCAN(DC) will indeed be able to accurately reproduce both structural and dynamical properties of liquid water over a wide range of temperatures and pressures. We are planning to report a more exhaustive analysis of the performance of MB-SCAN(DC) for liquid water in a future study.

2. Throughout the manuscript, there should be additional lines appearing in most plots, which are the MB-pol results. This would allow readers to judge the relative error compared to the best MB results. It would also be useful to have all the numbers in the plots tabulated in supp info.

3. The previous point is especially important in Figs 5 + 6 + 7. Also, how large are nuclear effects here? One should plot the MB-pol results in MD and PIMD, rather than simple comparison to experiment.

We thank the Reviewer for this suggestion. We added the results obtained from classical MD simulations in Figs. 5-7. Regarding nuclear quantum effects, as shown in Refs. 77 and 80 as well as in J. Chem. Phys. 147, 244504 (2017), nuclear quantum effects play a negligible role in determining the structural and dynamical properties of liquid water modeled with MB-pol. This is related to competing quantum effects along the stretching and bending coordinates which are discussed by Manolopoulos and coworkers in J. Chem. Phys. 131, 024501 (2009). In order to avoid making the figures too crowded, we added Fig. 10 to the Supplementary Information where we compare the MD and PIMD RDFs calculated with both MB-pol and MB-SCAN(DC) with the corresponding experimental RDFs.

4. Fig 4b suggests that MB-DC-SCAN gets the boats 1 and 2 energies in the wrong order, but one needs the actual numbers to tell.

The Reviewer is correct. DC-SCAN predicts the interaction energy for the bag isomer to be slightly higher than the corresponding value for the the cyclic isomer. We have added Tables S2-S5 with the actual values for each individual many-body contribution as well as the total interaction energies of the hexamer isomers to the the Supplementary Information.

Regarding the inversion between the bag and cyclic chair isomers, we believe that it is due to functional-driven errors associated with the SCAN functional. In Fig. 2 of Ref. 51, we show that SCAN indeed describes relatively better the two-dimensional isomers (cyclic chair and cyclic boat isomers) of the water hexamer than the three-dimensional isomers (prism and cage isomers), with the bag isomer effectively representing the “transition” between these two different trends.

5. An inset plotting errors for Fig 4(b) would help. Here DC-SCAN is worse than SCAN beyond about 2.1 Å. Why is that not a problem? Also, HF-DFT is supposed to yield accurate energies for stretched heterodimers, but it looks like the correction is vanishing at large R. Please explain.

We believe that the Reviewer is referring to Fig. 2b.

We have added an inset plot in Fig. 2b showing the errors of SCAN and DC-SCAN with respect to the CCSD(T)-F12 values at larger distances. It is known that SCAN includes intermediate-range dispersion interactions, but it lacks a proper description of the long-range component. Upon applying the density correction, which removes the (overbinding) 2B density-driven errors, the lack of long-range dispersion interactions results in DC-SCAN underbinding the water dimer at larger water–water distances. It follows that the reason behind SCAN giving slightly better results than DC-SCAN at long range is the (fortuitous) compensation in SCAN between density-driven errors (which overbind) and lack of long-range dispersion interactions.

It is worth mentioning here that the errors associated with SCAN and DC-SCAN converge asymptotically at

very long distances. We disregarded the inclusion of additive dispersion correction because from figure S1 it is apparent that all conventional dispersion corrections currently available significantly overbind the isomers of the water hexamer, resulting in poorer agreement with the CCSD(T)/CBS reference values than provided by the (dispersionless) DC-SCAN functional. In this regard, we believe that a proper treatment of the dispersion interactions with the DC-SCAN functional as a function of the water–water distance should include a correct description of the long-range component which smoothly transitions into the intermediate-range component intrinsic to the DC-SCAN functional as the water–water distance decreases. Work along these lines is currently ongoing in our group.

6. The only explanation I can see for the DC-SCAN0 result in Fig 1 is that the mixing of 25% exchange substantially worsens the energy error, and that 0% exchange is best. This is an important result (showing that SCAN0 is worse for water than SCAN, once evaluated on a good density). If the authors run 50% or even 100% exact exchange, they could confirm this.

We thank the Reviewer for this comment. It is indeed an important point and following Reviewer’s suggestion we have calculated the 2B energies for the first eight low-energy isomers of the water hexamer with SCAN_α ($\alpha = 0.50$), DC-SCAN_α ($\alpha = 0.50$), SCAN_α ($\alpha = 1.00$) and DC-SCAN_α ($\alpha = 1.00$), where α defines the fraction of Hartree-Fock exchange. The results, along with those obtained with the SCAN and SCAN0 functionals are shown in Fig. S2 of the Supplementary Information. Fig. S2 shows that, although density-driven errors decrease as the fraction of Hartree-Fock exchange increases, the agreement with the CCSD(T)/CBS reference values deteriorates. When evaluated on a good density the agreement with respect to the CCSD(T)/CBS reference values is: $\text{DC-SCAN} > \text{DC-SCAN0} > \text{DC-SCAN}_\alpha$ ($\alpha = 0.50$) $>$ DC-SCAN_α ($\alpha = 1.00$), with DC-SCAN providing the closest agreement.

7. At the bottom of page 25, the second last sentence contains “nonempirical” twice, when it should probably be ‘empirical’ the second time. But even then, it is too broad, as ωB97 and double hybrids yield more accurate densities.

We thank the Reviewer for noting this. We have revised the sentence as follows: “By many measures, the best nonempirical functionals predict more accurate densities for neutral atoms than the heavily-parameterized empirical functionals or even Hartree-Fock theory.”

8. It would be useful for the authors to include 1 or 2 other density functionals for comparison. For example, PBE should also improve with HF density, but presumably not as much as SCAN does. Also, how do these results compare with those of $\omega\text{B97M-V}$?

The Reviewer is completely correct. We reported a systematic analysis of density-driven errors in water for various exchange-correlation functionals in Ref. 105 and indeed found that DC-PBE performs significantly better than PBE, while the improvement due to the density-correction applied on $\omega\text{B97M-V}$ is minimal since the $\omega\text{B97M-V}$ density is already quite good. Since the analysis of various DC-DFT models for water is reported in Ref. 105 and in order to avoid to make the figures too crowded, we added Fig. 7 to the Supplementary Information with comparisons between self-consistent and density-corrected PBE-D3, SCAN, and $\omega\text{B97M-V}$ functionals for the water hexamer. Although from Ref. 105 it is apparent self-consistent $\omega\text{B97M-V}$ and other hybrids generate smaller density-driven errors compared to GGA and meta-GGA functionals, DC-SCAN outperforms DC- $\omega\text{B97M-V}$ when evaluated on a the HF density.

In this regard, we would like to mention that in Chem. Sci. 10, 8211 (2019), we developed a MB- $\omega\text{B97M-V}$ PEF derived from $\omega\text{B97M-V}$ 2B and 3B data and showed that it performs relatively well in MD simulations of liquid water, displaying only some minor deficiencies at the 3B level.

Response to Reviewer 3

The authors report a study of accuracy of density-corrected SCAN (DC-SCAN), a flavour of meta-GGA DFT, for the description of water. They benchmark DC-SCAN against coupled-cluster energies for water dimers and small water clusters, and assess its accuracy for thermodynamic properties of liquid water such as its density and O—O radial distribution function by comparing classical molecular dynamics (MD) simulations to experimental data. In order to render these MD simulations affordable they fit a force-field of MBpol architecture to DC-SCAN reference data, which is then used as a surrogate for first-principles DC-SCAN calculations.

Although the study is limited to a particular flavour of DFT and its accuracy for a single system — water — the detailed, yet clear, instructive, and convincing discussion of accuracy of DC-SCAN for describing the energetics of water dimers and clusters warrants publication. Nonetheless, there are areas in which the manuscript could be improved.

We thank the Reviewer for his/her/their careful reading and positive assessment of our manuscript. We address all comments below.

While the comparison of DC-SCAN to CCSD(T)-CBS for the dimers and clusters benefits from the ability to compare both levels of theory on an equal footing, since computing energies is tractable in both cases, the assessment of the accuracy of DC-SCAN for thermodynamic properties of water is more complicated for two main reasons:

(1) the computational cost associated with DC-SCAN forces the use of MB-SCAN(DC) as a surrogate in the MD simulations and simultaneously prevents direct assessment of how reproduction errors impact thermodynamic observables.

(2) since reference quantum chemistry data is even more unaffordable, experiment provides the best measure of the performance of the potential. However, comparison with experiment in principle requires rigorously accounting for thermal AND quantum fluctuations of the nuclei. Unfortunately, the authors only present results of classical MD simulations here. Indeed, both deuteration experiments and theoretical works such as www.pnas.org/cgi/doi/10.1073/pnas.1815117116 suggest quantum effects to increase the density of water at standard conditions (by of the order of 1%), promising even better agreement with experiment for MB-SCAN(DC) path-integral MD simulations (which should be rather affordable given the nature of the potential). Similarly deuteration experiments suggest that quantum effects lead to a slight de-structuring of the O—O RDF of water, which again promises improved agreement with experiment for MB-SCAN(DC) path-integral MD simulations, compared to classical simulations.

In my view, both points warrant further discussion in the manuscript.

We agree with the Reviewer that assessing the accuracy and reliability of a water model for bulk simulations is a challenging task. We also agree with the Reviewer that extensive comparisons with the experimental data are necessary for a complete assessment. However, we would also like to mention that, within a relatively small uncertainty, it is currently possible to also use the MB-pol model to assess the accuracy of other water models, which allows for directly connecting molecular interactions and experimental observables.

To place things in context, MB-pol has been extensively used by us and others to model the properties of water from small clusters in the gas phase (Science 2016, 351, 1310; Science 2016, 352, 1194; J. Chem. Phys. 2016, 145, 064308; J. Chem. Phys. 2016, 144, 061101; J. Am. Chem. Soc. 2017, 139, 7082; Phys. Chem. Chem. Phys. 2018, 20, 26809; J. Chem. Phys. 2018, 148, 234102; J. Chem Phys. 2018, 148, 124116; J. Chem. Phys. 2018, 148, 102303; J. Chem. Phys. 2018, 148, 084303; J. Phys. Chem. B 2019, 123, 9428; ACS Omega 2019, 4, 22581; Phys. Chem. Chem. Phys. 2020, 22, 1035) to bulk water (J. Chem. Theory Comput. 2015, 11, 1145; J. Chem. Theory Comput. 2016, 12, 1953; J. Chem. Phys. 2017, 147, 244504; Phys. Rev. Lett. 2018, 121, 137401; J. Phys. Chem. B 2018, 122, 10754; Nat. Commun. 2018, 9, 247), the air/water interface (J. Am. Chem. Soc. 2016, 138, 3912; J. Phys. Chem. B 2018, 122, 4356; J. Phys. Chem. A 2018, 122, 18, 4457; J. Phys. Chem. Lett. 2018, 9, 6744; Phys. Rev. Lett. 2018, 121, 246101), and ice (J. Chem. Theory. Comp. 2017, 13, 1778; J. Phys. Chem. Lett. 2017, 8, 2579; J. Phys. Chem. B 2018, 122, 10572).

Importantly, MB-pol has also been used as a reference for the assessment of the accuracy of modern polarizable models (e.g., see J. Chem. Theory Comput. 2019, 15, 5001 and J. Phys. Chem. Lett. 2020, 11, 419) as well as to improve exchange-correlation functionals for water (e.g., see J. Chem. Phys. 2016, 144, 224101; J. Chem. Phys. 2019, 151, 144102). We feel that it is reasonable to say that MB-pol is currently recognized by the community as the prototypical water model as demonstrated by WIREs Wiley Interdiscip. Rev. Comput. Mol. Sci. 2018, 8, e1355 where Demerdash, Wang, and Head-Gordon state that “At present, MB-pol achieves unprecedented accuracy in describing water properties from the dimer to the condensed phase and is perhaps one of the all-around best MM water models”. Based on the demonstrated “realism” of MB-pol, we believe that MB-SCAN(DC) provides a similarly accurate description of water from the gas to the liquid phase because it shares with MB-pol all the key features that we found important to guarantee accuracy and transferability in a molecular model of water.

In this context, our analyses with MB-pol reported in Refs. 77 and 80 as well as in J. Chem. Phys. 147, 244504 (2017) show that nuclear quantum effects play a negligible role in determining both structural and dynamical properties of liquid water at ambient conditions. In the end, the difference in the melting point between H₂O and D₂O is only 3.8 °C. This is related to competing quantum effects along the stretching and

bending coordinates which are discussed by Manolopoulos and coworkers in *J. Chem. Phys.* 131, 024501 (2009). This is the reason why our initial MD simulations were carried out at the classical level. During the review process, we also performed PIMD simulations at ambient conditions with MB-SCAN(DC). However, in order to avoid making the figures too crowded, we added Fig. 10 to the Supplementary Information where we compare the MD and PIMD RDFs calculated with both MB-pol and MB-SCAN(DC) with the corresponding experimental results. As expected from the MB-pol results, the differences in the MD and PIMD O-O RDFs are minimal, with only small differences appearing in the O-H and H-H RDFs due to zero-point energy effects.

This allows us to briefly comment on the results reported by Cheng et al. in *Proc. Natl. Acad. Sci. U.S.A.* 116, 1110 (2019) which are mentioned by the Reviewer. As shown in *Chem. Sci.* 10, 8211 (2019), revPBE0-D3, which is the exchange-correlation functional used by Cheng et al. to train the neural network that was then employed in the actual MD and PIMD simulations, suffers from noticeable 2B and 3B errors that fortuitously compensate. This results in an incorrect description of the local "curvature" of the underlying energy landscape, which is directly related to the "magnitude" of nuclear quantum effects predicted by revPBE0-D3. It should be noted that this error compensation is observed in many exchange-correlation functionals commonly applied to water [e.g., see Fig. 8 of *Chem. Rev.* 116, 7501 (2016)]. Importantly, we showed in Ref. 105 that these errors in common exchange-correlation functionals are both functional-driven and density-driven in nature. We believe that taking these limitations into account makes the results of *Proc. Natl. Acad. Sci. U.S.A.* 116, 1110 (2019) somewhat less reliable. In this context, it should be mentioned that PIMD simulations with MB-pol at ambient pressure predict a lower density of liquid water compared to analogous MD simulations (Ref. 80). As expected, the PIMD simulations with MB-SCAN(DC) indeed predict a density of 0.984 g/cm³, which is slightly lower than that obtained from the analogous MD simulations (0.986 g/cm³).

A few additional points are worth mentioning:

(1) In my view the abstract overreaches slightly by claiming that the work "demonstrates that MB-SCAN(DC) is effectively the first DFT-based model that correctly describes water from the gas to the condensed phase", given that the condensed phase includes the full phase-diagram with all different crystalline forms and their diverse anomalous behaviour, some of which is driven by quantum-mechanical effects.

We understand the point made by the Reviewer. We based our conclusions on the similarity between MB-SCAN(DC) and MB-pol. Since MB-pol has been shown to describe the ice phases quite well [*J. Chem. Theory Comput.* 213, 1778 (2017); *J. Phys. Chem. Lett.* 8, 2579 (2017); *J. Phys. Chem. B* 122, 10572 (2018)], we expect MB-SCAN(DC) to provide similar accuracy. To avoid confusion, we modified our sentence in: "demonstrates that MB-SCAN(DC) is effectively the first DFT-based model that correctly describes water from the gas to the liquid phase".

(2) With respect to Fig.1, a comment regarding the systematic offset and the comparatively very small errors in relative energy would be appreciated.

At the end of the caption of Fig. 1, we have added: "The 2-body energy, on average, contributes ~80-85% to the total interaction energy in water.¹⁴ The errors associated with a given functional relative to the CCSD(T)/CBS values are roughly the same for each isomer."

(3) Fig.2 supposedly shows that "DC-SCAN exhibits significantly smaller errors compared to SCAN for all dimers, independent of the O...O distance". While the size of the markers used in panel (a) render assessing this difficult, panel (b) suggests this to be untrue above around 4.3 Angstrom, although the errors are very small at that point.

To clarify this point, we have added an inset plot in Fig. 2b showing the errors of SCAN and DC-SCAN with respect to the CCSD(T)-F12 values at larger distances. It is known that SCAN includes intermediate-range dispersion interactions, but it lacks a proper description of the long-range component. Upon applying the density correction, which removes the (overbinding) 2B density-driven errors, the lack of long-range dispersion interactions results in DC-SCAN underbinding the water dimer at larger water-water distances. It follows that the reason behind SCAN giving slightly better results than DC-SCAN at long range is the (fortuitous) compensation in SCAN between density-driven errors (which overbind) and lack of long-range dispersion interactions.

It is worth mentioning that the errors associated with SCAN and DC-SCAN converge asymptotically at very long range. We disregarded the inclusion of additive dispersion correction because from figure S1 it is apparent that all conventional dispersion corrections currently available significantly overbind the isomers of the water hexamer, resulting in poorer agreement with the CCSD(T)/CBS reference values than provided by the (dispersionless) DC-SCAN functional. In this regard, we believe that a proper treatment of the dispersion interactions with the DC-SCAN functional as a function of the water-water distance should include a correct description of the long-range component which smoothly transitions into the intermediate-range component intrinsic to the DC-SCAN functional as the water-water distance decreases. Work along these lines is currently ongoing in our group.

(4) A brief outline of the many-body expansion underlying the construction of the MB-SCAN(DC) potential would be appreciated.

We thank the Reviewer for pointing this out. We rearranged the Results section to include an initial subsection ("Theoretical background") describing some details of the many-body expansion which were originally discussed in the Methods section. We believe that describing the key components of the MB-SCAN(DC) PEF before reporting the actual results would help the reader better follow the flow of our manuscript.

(5) It remains unclear what configurations (and associated energies) underlie the MB-SCAN(DC) potential and why these could not have been computed using the gold standard, i.e. CCSD(T)-CBS, directly, given that CCSD(T)-CBS calculations for dimers and small clusters are clearly affordable. On this note, the authors might want to highlight that a key benefit of the MB-SCAN(DC) potential over a hypothetical MB-CCSD(T)-CBS potential is that it provides evidence of accuracy of DC-SCAN, which in turn is transferable and hopefully accurate for other systems, for which CCSD(T)-CBS calculations are not viable.

As described in the Methods section, the 1B term of MB-SCAN(DC) is represented by the Partridge-Schwenke PEF (Ref. 115), while the 2B and 3B terms are fitted to reproduce 2B and 3B energies calculated with DC-SCAN for the same training sets of water dimers and trimers used in the development of MB-pol (Refs. 78 and 79). All higher-body terms are described by classical polarization as in MB-pol.

In this regard, it should be noted that an MB-CCSD(T)-CBS potential actually exists and is indeed MB-pol, which was trained on CCSD(T)/CBS 2B and 3B data (Refs. 78 and 79) and shown to also accurately reproduce CCSD(T) higher-body interactions (Ref. 77). While we have demonstrated in previous studies that MB-pol provides a very accurate description of water, we believe that the significantly lower computational cost associated with DC-SCAN calculations can enable the routine development of data-driven many-body MB-SCAN(DC) PEFs for generic molecules which are trained on DC-SCAN data but effectively display CCSD(T) accuracy. We believe that this is an important result.

Besides the implications for MD simulations of generic molecules from the gas to the liquid phase, our study also points out that density-driven errors in SCAN applied to water are significant and may be the reason for the somewhat erratic behavior observed in *ab initio* MD simulations of liquid water with both SCAN (and other SCAN variants) and neural networks trained on SCAN data, with some properties being reproduced relatively well and other properties being appreciably different from the experimental data (e.g., Proc. Natl. Acad. Sci. USA 2017, 114, 10846; Proc. Natl. Acad. Sci. U.S.A. 2020, 117, 26040; J. Chem. Theory Comput. 2021, 17, 3065; arXiv:2104.14410).

Finally, our study suggests that our many-body formalism provides a data-driven approach distinct from neural networks which, consistently building upon a rigorous formulation of density-corrected energies, enables significantly more accurate simulations of liquid water compared to neural networks trained on *ab initio* DFT-based simulations of liquid water. Given the widespread use of neural network potentials trained on DFT data, we believe that our study is particularly timely and important for the community interested in developing accurate data-driven models.

(6) If the authors have investigated the errors in the thermodynamic observables (with respect to the first-principles reference) arising from the MB form/fitting, this would be an interesting addition to the manuscript. In particular, in view of the hypothesis that the reduced density errors in the DC-SCAN reference data facilitate the fitting.

This is an important point. Although we were originally planning to report a more extensive analysis of the phase behavior of water using MB-SCAN(DC) in a future study, we added Figs. 11 and 12 to the Supplementary Information showing the temperature dependence of the enthalpy of vaporization and isothermal compressibility. These figures indeed show that MB-SCAN(DC) achieves an accuracy similar to that of MB-pol. It should be noted that both the enthalpy of vaporization and isothermal compressibility predicted by MB-SCAN(DC) are consistent with the analyses of many-body energies and interactions with DC-SCAN. In particular, the small underestimation of the water dimer interaction energies leads to smaller enthalpies of vaporization, which are also reflected in lower densities compared to experiment. These differences consistently also result in higher values for the isothermal compressibility. In this regard, it should be noted that the temperature dependence of the isothermal compressibility predicted by MB-SCAN(DC) is in significantly closer agreement with the experimental values compared to the results obtained with a deep neural network trained on SCAN in Proc. Natl. Acad. Sci. U.S.A. 117, 26040-26046 (2020).

Finally, the first half of the discussion appears to effectively reiterate the introduction. Personally, I feel that this repetition is redundant.

We thank the Reviewer for pointing this out. We removed the second paragraph of the Discussion which we believe improves the overall flow of our manuscript.

REVIEWERS' COMMENTS

Reviewer #2 (Remarks to the Author):

The authors have responded satisfactorily to all issues raised in my previous report, and I now recommend the paper for publication.

Reviewer #3 (Remarks to the Author):

The authors have convincingly addressed my concerns, clarifying all key points. In my view the revised manuscript comprises interesting and very detailed research, which is communicated well, and could be published in its current form.

I would like to add two suggestions, regarding minor changes to the presentation, which the authors may choose to adopt or reject as they prefer:

(1) in my view Fig.4 would benefit from insets showing the errors relative to the CCSD(T) reference, and

(2) a sentence discussing the much larger errorbars for the MB-SCAN(DC) self-diffusion coefficients at 320 and 340K in Fig.7 would be appreciated.

Response to Reviewer 2

The authors have responded satisfactorily to all issues raised in my previous report, and I now recommend the paper for publication.

We thank the Reviewer for careful reading of our revised manuscript, and his/her/their support for publication.

Response to Reviewer 3

The authors have convincingly addressed my concerns, clarifying all key points. In my view the revised manuscript comprises interesting and very detailed research, which is communicated well, and could be published in its current form.

We thank the Reviewer for careful reading of our revised manuscript, and his/her/their support for publication.

I would like to add two suggestions, regarding minor changes to the presentation, which the authors may choose to adopt or reject as they prefer: (1) in my view Fig.4 would benefit from insets showing the errors relative to the CCSD(T) reference

We thank the Reviewer for this suggestion. We tried to include the errors in Fig. 4 but felt that it made the figure too crowded. Therefore, we decided to show the errors in a separate figure of the Supplementary Information (Fig. 7 in the revised version) and modified the manuscript accordingly to refer to the new supplementary figure.

(2) a sentence discussing the much larger errorbars for the MB-SCAN(DC) self-diffusion coefficients at 320 and 340K in Fig.7 would be appreciated.

We thank the Reviewer for this suggestion. We have added a sentence to the main text describing the origin of the larger error bars at high temperature.